# Object Play and Autism Spectrum Disorders Symptoms in Toddlers Aged 12–37 Month at Risk of Developmental Disorders

**DOI:** 10.3390/children10060995

**Published:** 2023-06-01

**Authors:** Rafał Kawa

**Affiliations:** Faculty of Psychology, University of Warsaw, 00-183 Warsaw, Poland; rkawa@psych.uw.edu.pl; Tel.: +48-608520636

**Keywords:** object play, exploratory activity, risk of developmental disorders, autism spectrum disorders

## Abstract

Various studies have shown a relationship between the development of language and object play in children. Children with Autism spectrum disorder (ASD) show difficulties in both of these areas of functioning. But the knowledge about the relationships between the severity of ASD symptoms and object play in children at risk of developmental disorders remains limited. To explore these relationships, 44 children aged 13–37 months took part in this study. Object play and ASD symptoms were assessed in two age groups (13–18 months and 24–37 months). The results show that ASD symptom intensity is related to object play complexity in children at risk of developmental disorders. However, these relationships are different depending on the age of the children.

## 1. Introduction

Object play is a skill that emerges early in a child’s development. It is closely linked with the development of cognitive, language, motor, and visual skills [1]. Object play develops gradually. One of its first forms is exploratory play or sensorimotor play, which usually involves manipulations of a single object. The primary goal of this type of play is to learn about the physical world [2]. Exploratory play transitions into relational play, which involves more complex manipulations of two or more objects [3]. Another form of object play is functional play, which includes the functions of objects in children’s play activities, with objects being used in accordance with their purposes. For example, the child uses a toy hammer to drive toy nails or uses a toy hair dryer. Functional play usually appears in the play repertoire at approximately 12 months of age [4]. Between 18 and 24 months of age, children develop symbolic play. This type of play is more complex and qualitatively different from play activities in preceding forms. It involves symbolic thinking and mental manipulation of objects. One kind of symbolic play is pretend play [5]. The development of object play is not hierarchical, and the kinds of play typical for a given age may be observed at other stages of development. However, object play becomes more complex over time, building on the elements of previous stages [6]. This is why, given the development of functional and symbolic play, when comparing the way children play under 18 months and at over 24 months of age, we can expect significant differences in terms of form and complexity.

A number of studies have demonstrated a relationship between the development of language, symbolic skills, and object play, especially pretend play [7]. Positive correlations were found between language development and symbolic play in children [8]. Those who demonstrated more advanced symbolic play skills and presented more complex and sophisticated forms of object play were more likely to exhibit higher levels of language skills.

Children with autism spectrum disorder (ASD) and children at high risk for ASD differ from typically developing children and children with other developmental disorders with respect to object play [9]. Children with ASD present repetitive patterns of behavior and difficulties in social communication, which are the core symptoms of ASD according to the Diagnostic and Statistical Manual of Mental Disorders [10] criteria. Repetitive patterns of behavior define a broad spectrum of behaviors characterized by sameness, rigidity, and repetitiveness. The presence of repetitive behavior has an influence on the development of play skills in children with ASD [11]. For example, Libby et al. [12] argue that preoccupation with sensorimotor play might suppress the emergence of functional and symbolic skills. However, even though repetitive behaviors belong to the core symptoms of ASD, it should be noted that they also tend to be present in typically developing children [13,14] and children with other developmental disorders.

Research has shown that children with ASD are more likely to play in a more stereotyped, less varied, and less flexible way than their typically developing peers [15]. Moreover, it has been noted that they tend to engage in more exploratory play and in stimulating activities compared with typically developing children and children with other developmental disorders [2]. Children on the spectrum are also less likely to become involved in relational play, and their play is less complex than their typically developing counterparts [12]. Studies have also shown that children with ASD experience difficulties engaging in more complex play behavior (e.g., pretend play). In comparison to typically developing children, children with ASD often lack the ability to present original ideas for pretend play, and their play often seems rote and less spontaneous [3].

Untypical object play has been observed not only in children with ASD but also those diagnosed with ASD later in life and in siblings at risk of ASD [16]. It has been noted that children at high risk of ASD are more likely to engage in less complex exploratory play, e.g., put objects in the mouth more often, demonstrate excessive visual fixation, present repetitive visual fixation, engage in the excessive spinning of objects versus children at low risk of ASD [17,18,19]. In addition, children at high risk of ASD have been observed to present a play that was more repetitive and less functional [15].

Even though a lot of studies addressed the issue of object play in children at high risk of ASD, as well as the methods of measuring object play in children [9], our knowledge about the relationships between the severity of ASD symptoms and object play in children at risk of developmental disorders remains limited. Furthermore, studies of children at high risk of ASD mostly enroll participants less than 24 months of age and use objects familiar to children, such as popular toys. There is still insufficient data on the relationships between object play and ASD symptoms, either in children aged less than 18 months or more than 24 months. It is entirely possible that these relationships are somewhat different in both younger and older populations due to the development of symbolic skills. The purpose of this study was to investigate the relationships between object play and the severity of ASD symptoms in younger children aged 12-18 months and older children aged 24–36 months at high risk of developmental disorders. The study aimed to find answers to the following research questions: (1) How is object play related to ASD symptoms in children at risk of developmental disorders? (2) Are the relationships between object play and ASD symptoms different in younger and older children at risk of developmental disorders?

## 2. Materials and Methods

### 2.1. Participants

Forty-four children at risk of developmental disorders aged 13–37 months (M = 22.2; SD = 8.77) took part in the study. The sample consisted of 10 girls and 34 boys and had a wide range of ages in participants; therefore, the children were split into two groups. In the younger group, there were 23 children aged 13–18 months (M = 14.75; SD = 1.42), 4 of them girls and 19 boys. In the older group, there were 21 children aged 24–37 months (M = 30; SD = 5.86). The group consisted of 4 girls and 17 boys. Table 1 shows the population description and ADOS-2 scores. Overrepresentation of boys is often reported when children at risk of developmental disorders are studied [20].

### 2.2. Procedure

To recruit participants into the group, ads were posted on social media and on internet websites for parents of children under 3 years of age. This method recruited 6 children from the Warsaw region (Poland) into the study. Other children were recruited through contact with legal guardians of participants that took part in a previous research project. In this study, the First Year Inventory ([21], unpublished Polish version–Kawa, Pisula, Tomalski, Słowińska, 2015) was used to identify children at risk of developmental disorders. Parents of all participants filled out the FYI at the age of the child of 12 months (±2 weeks). The study enrolled children aged 11.5 and 12.5 months. In the current study, besides age (11–37 months), the inclusion criteria concerned term of birth (38 to 42 weeks), FYI score above 13 points, and no diagnosis of developmental, genetic, or neurological disorders at the time of entry. The threshold of FYI score of 13 was selected based on data collected using that scale from parents of typically developing children, children with developmental disorders, and children with ASD [22]. Object play measurements and ASD symptoms assessments were conducted no more than 3 weeks apart. The legal guardians were asked to provide consent for participation in this study. The recruitment method was described in more detail in a previous paper [23].

### 2.3. Measures

#### 2.3.1. First-Year Inventory

The FYI [21] is a questionnaire consisting of 63 items. The tool has norms to identify 12-month-old infants who are likely to acquire an ASD diagnosis in the future. The FYI assesses behavior in two main developmental categories: social communication and sensory-regulatory systems. Social orienting and receptive communication, social-affective engagement, imitation, expressive communication, sensory processing, regulatory patterns, reactivity, and repetitive behavior are the eight particular characteristics that constitute the two domains. A quasi-logarithmic risk score of 0 to 50 points is utilized for each of the eight components as well as the two domains. The sensory-regulatory and social-communication domain scores are averaged to provide the overall FYI risk score. Greater scores suggest that the child’s parent reported more unusual behaviors. Cronbach’s alpha for the US version is 0.81, 0.71 for the social-communication domain, and 0.63 for the sensory-regulatory domain.

The Polish version of FYI was first translated by a team of 3 psychologists-diagnosticians. Then, the translated versions were compared, and a unified version was prepared, which was corrected by a native speaker. Subsequently, back-translation was carried out and sent to the authors of the tool to verify the translation. The Polish version of FYI [21] contains the same number of questions as the original version. Cronbach’s alpha for the Polish version was 0.72. After the translation was accepted by the authors of the English version, data collection began.

The study also used a demographic survey in which parents were asked about their age, education, current occupation, siblings of the examined child, the month of delivery, the child’s disorders, the child’s medication intake, or the child’s attendance at therapeutic/rehabilitation classes.

#### 2.3.2. Object Play Measurement Sessions

To measure object play, experimental objects were designed and manufactured [23]. Each object measured 10 cm × 10 cm × 10 cm and was made from blue and red plastic, printed on a 3D printer (Figure 1). Each block had 6 electronic displays capable of showing animations. The animations were designed for the needs of this study. Each display showed an identical, simple animation when the objects were separate (e.g., a swimming fish). Once objects were connected, the displayed animations changed so that screens were merged into one. The more objects were connected together, the complexity of the animations increased. During a measurement session, children had 4 objects equipped with magnets to connect them.

Each child participated in one session. Object play sessions were conducted in the experimental room equipped with HD video cameras.

Each measurement lasted for six minutes. Measurement sessions in studies on object play in children of comparable ages [24] are typically brief, lasting between 5 and 15 min. The youngster, together with the investigator and parent, was present in the experimental room for the session. The caregiver was instructed to sit on the floor next to the child and observe him or her playing before the experimenter began taking measurements. The researcher then gave a demonstration, sitting in front of the child while connecting the blocks in three different ways—creating a row, a tower, or an L-shape. During the experimenter’s demonstration and trials, the youngster received no verbal instructions. The experimenter then placed the objects in front of the youngster, and the measurement began. Given the goal of the study, the analysis took into account the components of children’s play that entailed manipulating experimental items and allowed for the assessment of play complexity. Two HD video cameras were used to capture measurement sessions. In order to increase dependability and ensure that the child’s activity would be captured even if she was facing away from one of the cameras, two cameras were employed.

The partial interval approach was used to code the recording following each measurement by a person who was not involved in the session [25]. Each interval lasted for 10 s. The coding method involved two persons. There was 93% intra-rater reliability. The variables coded in this study are displayed in Table 2.

#### 2.3.3. Autism Spectrum Disorder Measurement

To assess the level of ASD symptoms in participants, the Autism Diagnostic Observation Schedule, Second Edition–(ADOS-2) [26] was used. It is a semi-structured, standardized assessment of communication, social interaction, play, and restricted and repetitive behaviors. The measure presents various activities that elicit behaviors directly related to the diagnosis of ASD. By observing and coding behaviors, data that informs diagnosis, treatment planning, and educational placement can be obtained. The ADOS-2 includes five modules, each requiring only 40 to 60 min to administer. The individual being evaluated is given only one module, selected on the basis of his or her expressive language level and chronological age. The tool consists of two scales: Social Affect, which measures difficulties in social communication, and Limited and Repeated behavior, which measures repetitive patterns of behavior. Module-T and Module 1 were used in this study. A Polish version is available [27].

## 3. Results

To obtain answers for the research questions, data analysis was based on Spearman’s rho partial correlation coefficients. Next, the significance of correlation coefficients was compared between the younger and older children’s groups using the z-statistics. Correlations were calculated separately for the group of children aged 13–18 months and 24–37 months.

Table 3 present the percentage of intervals with observed behavior. In the younger group, the mean ADOS total was 4.04 (SD = 3.02), in the Social Affect scale 3.83 (SD = 2.95), and 0.41 (SD = 0.20) in the Limited and Repeated behavior scale. The mean ADOS total score in the older group was 4.53 (SD = 3.03), in the Social Affect scale 3.08 (SD = 2.98), and 0.71 (SD = 0.70) in the Limited and Repeated behavior scale.

Table 4 presents the values of Spearman’s rho partial correlation coefficients obtained in the younger group of children aged 13–18 months.

In the younger group, the analyses showed moderate positive correlations between the ADOS total score and putting three objects together in various configurations and putting four objects together in various configurations. Positive correlations between the ADOS Social Affect score and putting three objects together in various configurations and putting four objects together in various configurations were also observed. Low positive correlations have been observed between the ADOS total score, ADOS Social Affect score, and putting two objects one on top of the other. Low negative correlations have been observed between ADOS total score, ADOS Social Affect score, and putting two objects together in a row. Nevertheless, statistically significant correlations have been observed only between connecting three and four objects and ADOS measures.

In the older group (Table 5), low positive correlations were found between ADOS total score, ADOS Social Affect score, and looking towards the object and hitting the object. Between ADOS Repeated Behavior score and rotating the object, moderate correlations were found. However, no correlations in the older group were statistically significant.

Next, the significance of correlation coefficients was compared between the younger and older children’s groups using the z-statistics. The results of these analyses are shown in Table 6.

Statistically significant differences between the correlation coefficients have been observed regarding looking at objects (negative correlations in the younger group), putting two objects in a row (negative correlations in the younger group), and connecting three objects in any configuration (positive correlations in the younger group) in ASD Total Score and Social Affect.

## 4. Discussion

The aim of the present study was to analyze the relationships between object play and ASD symptoms in children at risk for developmental disorders. For that purpose, we analyzed correlations between the severity of ASD symptoms and object play using novel objects designed to encourage children to manipulate and combine them in various configurations. The analyses were conducted separately in younger (13–18 months) and older participants (24–37 months).

The expectation was that the severity of ASD symptoms would be related to the complexity of play in the children at risk for developmental disorders. The results partially confirm this hypothesis. In addition, it was noted that these relationships are different in younger and older children. Numerous studies on children with ASD have found that they are more likely to engage in exploratory play and stimulating activities [3]. In contrast to typically developing peers, their play is usually more restricted and repetitive [17,18,19]. Some authors have linked this fact with the presence of stereotyped behaviors in children with ASD [12]. Exploratory play and stimulating activities often involve the manipulation of single objects [3]. It could, therefore, be expected that in this study as well higher severity of ASD symptoms would correlate with a greater frequency of manipulating single objects. However, the relationships found in the study group between ASD symptom severity and single-object manipulation, such as turning an object around its axis, were weak. Much stronger, positive correlations were obtained between the severity of ASD symptoms and combining objects and more complex play behaviors. They were the strongest when the child was manipulating three or four objects. In other words, younger children demonstrating more social function problems engaged more often in relational play and manipulated several objects combining them into various structures. This stays in opposition to some studies regarding object play in children with ASD [28]. Another notable fact was that these relationships were found between object play and ADOS total score and ADOS SA score, but not between object play and ADOS RRB score. We might expect [12] that the severity of stereotyped behavior would demonstrate the strongest relationships with object play. Our findings suggest otherwise, at least in the case of play of children at risk of developmental disorders.

## 5. Conclusions

One reason for the actual pattern of these relationships may be the fact that children at risk of developmental disorders whose social functioning is more impaired probably engage in relational play more often and initiate social interaction less often than typically developing children [18,19]. It should be noted, however, that in other studies, children with ASD often demonstrated difficulties with respect to relational play, and their play was usually less complex than that of their typically developing peers [12]. Still, the present study enrolled children not diagnosed with ASD, with ADOS-2 scores below the cut-off point, and thus the severity of symptoms was in all likelihood lower than in children with ASD diagnosis. However, we cannot conclude that children in the study who demonstrated greater severity of symptoms in the SA subscale were less likely to engage in social interaction in the present study since there was no control for social interaction.

On the other hand, the results may have been affected by the characteristics of objects used in the present study. Studies on play activity in children differ to a large degree regarding methodology and objects used in the observation of play behavior, which may have a relevant impact on the results [7]. Their design motivated children to combine them in various configurations. In addition, the procedure required that neither the experimenter nor the parent initiates interactions with the child. Consequently, younger children may have been more highly motivated to engage in that type of play, especially those children that demonstrate social function problems.

In turn, the relationships between object play and autism symptom severity observed in the older group were not statistically significant. The associations between relational play, i.e., manipulation of multiple objects, and ASD symptoms were very weak. Stronger relationships in this group were found between the variables associated with looking toward objects and the manipulation of a single object. It is worth pointing out that a comparison of the correlation coefficient has shown significant differences between the younger and older children regarding looking at objects, putting two objects in a row, connecting three and four objects in any configuration, and ASD Total Score and Social Affect.

These findings suggest that the relationships between the severity of ASD symptoms and object play may differ depending on children’s age. Perhaps older children, who would be expected to possess higher levels of symbolic skills and engage in more pretend play, were less motivated to engage in relational play. However, the symbolic play was not controlled for in the present study. On the other hand, the design of objects used in the study was not meant to motivate children to engage in symbolic play.

One of the limitations of the present study was the fact that social interactions between the child and experimenter, and caregiver were not measured. It would be a good idea to take into account this aspect of play when studying children of various ages and analyzing relationships between ASD symptoms and object play, given that social communication is one of the core symptoms of ASD. Another limitation was that the study did not control for the child’s mental age/level of development. This factor may have significantly affected the complexity of the participants’ play.

## Figures and Tables

**Figure 1 children-10-00995-f001:**
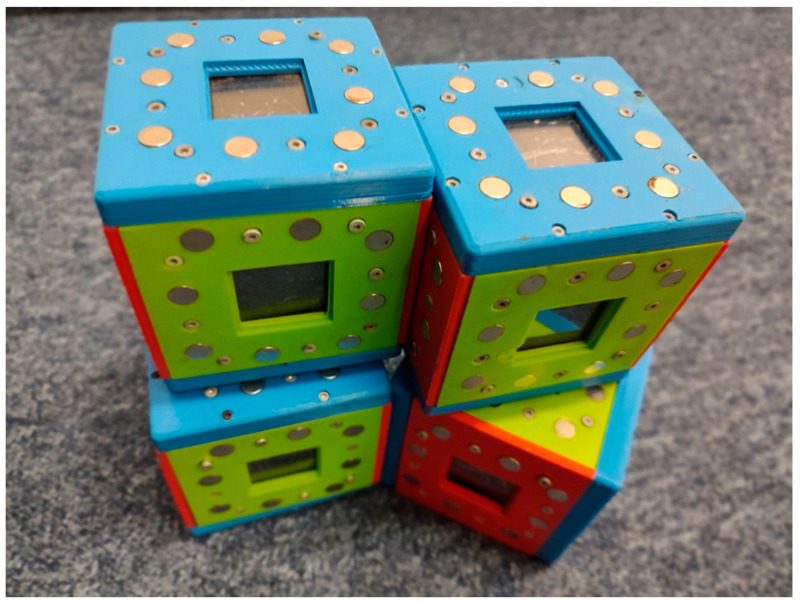
Objects used to measure object play.

**Table 1 children-10-00995-t001:** Population description and ADOS-2 scores measured in this study.

Variable	Younger Group (13–18 Months)	Older Group (24–37 Months)
Sample size	23 (4 girls and 19 boys)	21 (4 girls and 17 boys)
Age	M = 14.75; SD = 1.42	M = 30; SD = 5.86
ADOS-2 Total score	M = 4.04; SD = 3.02	M = 4.53; SD = 3.03
ADOS-2 SA	M = 3.83; SD = 2.95	M = 3.08; SD = 2.98
ADOS-2 LR	M = 0.41; SD = 0.20	M = 0.71; SD = 0.70

**Table 2 children-10-00995-t002:** Description of variables measured in this study.

Variable	Description
looking toward the object	the child directs gaze toward the object without touching the object
rotating the object	the child turns the object around its axis
hitting the object	the child hits the object with an open or closed palm or with another object
putting 2 objects together in a row	the child moves one object to another, they become connected, and the displayed animations change
putting 3 objects together in a row	the child moves one object towards two already connected objects, it becomes attached, and the displayed animations change
putting 4 objects together in a row	the child moves one object towards three already connected objects, it becomes attached, and the displayed animations change
putting 2 objects one on top of the other	the child places one object on top of another; they become connected, and the displayed animations change
putting 3 objects together in various configurations	the child moves one object towards two already connected objects, it becomes attached, and the displayed animations change
putting 4 objects together in various configurations	the child moves one object towards three already connected objects, it becomes attached, and the displayed animations change

**Table 3 children-10-00995-t003:** Percentage of intervals with observed behavior.

Exploratory Activity	Younger Group (13–18 Months)	Older Group (24–37 Months)
looking towards objects	86.29%	88%
rotating objects	20.16%	19.81%
hitting objects	5.04%	5.09%
putting 2 objects in a row	6.25%	7.54%
putting 2 objects one on top of the other	3.45%	3%
connecting 3 objects in any configuration	3.16%	4.45%
combining 4 objects in any configuration	2.37%	2.45%

**Table 4 children-10-00995-t004:** Spearman’s rho partial correlation coefficients obtained in the younger group of children aged 13–18 months.

Object Play		ADOS Total Score	ADOS Limited and Repeated Behavior	ADOS Social Affect
looking towards objects	Correlation Coefficient	−0.220	0.342	−0.296
	Sig. (2-tailed)	0.301	0.102	0.160
rotating objects	Correlation Coefficient	0.200	0.319	0.206
	Sig. (2-tailed)	0.348	0.128	0.333
hitting objects	Correlation Coefficient	−0.115	0.382	−0.195
	Sig. (2-tailed)	0.594	0.065	0.362
putting 2 objects in a row	Correlation Coefficient	−0.369	−0.097	−0.353
	Sig. (2-tailed)	0.076	0.651	0.090
putting 2 objects one on top of the other	Correlation Coefficient	0.355	−0.119	0.383
	Sig. (2-tailed)	0.088	0.578	0.065
connecting 3 objects in any configuration	Correlation Coefficient	0.480	0.124	0.501
	Sig. (2-tailed)	0.018	0.564	0.013
combining 4 objects in any configuration	Correlation Coefficient	0.464	0.083	0.493
	Sig. (2-tailed)	0.023	0.700	0.014

**Table 5 children-10-00995-t005:** Spearman’s rho partial correlation coefficients acquired in the older group of children aged 24–37 months.

Object Play		ADOS Total Score	ADOS Limited and Repeated Behavior	ADOS Social Affect
looking towards objects	Correlation Coefficient	0.376	0.152	0.393
	Sig. (2-tailed)	0.093	0.511	0.078
rotating objects	Correlation Coefficient	0.007	0.388	−0.087
	Sig. (2-tailed)	0.976	0.082	0.707
hitting objects	Correlation Coefficient	0.343	0.397	0.290
	Sig. (2-tailed)	0.128	0.075	0.202
putting 2 objects in a row	Correlation Coefficient	−0.077	0.238	−0.126
	Sig. (2-tailed)	0.740	0.299	0.587
putting 2 objects one on top of the other	Correlation Coefficient	−0.115	0.108	−0.118
	Sig. (2-tailed)	0.619	0.642	0.611
connecting 3 objects in any configuration	Correlation Coefficient	−0.256	−0.222	−0.169
	Sig. (2-tailed)	0.263	0.333	0.463
combining 4 objects in any configuration	Correlation Coefficient	0.046	−0.110	0.046
	Sig. (2-tailed)	0.843	0.635	0.842

**Table 6 children-10-00995-t006:** Comparison of significance levels of correlation coefficients in the younger and older group.

Object Play	ADOS Total Score	ADOS Limited and Repeated Behavior	ADOS Social Affect
looking towards objects	0.05	0.53	0.02
rotating objects	0.54	0.80	0.36
hitting objects	0.14	0.95	0.12
putting 2 objects in a row	0.29	0.29	0.45
putting 2 objects one on top of the other	0.13	0.48	0.10
connecting 3 objects in any configuration	0.01	0.28	0.02
combining 4 objects in any configuration	0.16	0.55	0.12

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
