# Peer review of "Object Play and Autism Spectrum Disorders Symptoms in Toddlers Aged 12–37 Month at Risk of Developmental Disorders"

_children, 2023, doi:10.3390/children10060995_

Round 1
Reviewer 1 Report
1. Any psychometric evidence for the ADOS -2 in its polish version? e.g., reliability, validity, sensitivity, specificity?
2. I think in the methods section, there should be a subsection describing how the data would be analyzed (i.e., how different statistical methods were chosen to answer certain research questions). It might be also good to list specific research questions the author(s) would like to answer in the introduction section.
3. I think the key concern for me is that I am not sure if the author(s) have been able to bring the findings of this study to a wider context. That is, I am not sure if the findings have been discussed to a level that could really add to the existing literature.
Author Response
1. ADOS-2 has been translated and adapted to the Polish culture (Chojnicka & Pisula, 2018). The authors of the adaptation recruited 401 participants: 193 with ASDs (ASD group) and 78 with non-spectrum disorders, plus 130 typically developing participants (control group). ADOS-2-PL was found to have high interrater reliability, internal consistency and test–retest reliability. Confirmatory factor analysis confirmed a good fit of the Polish data to the two-factor model of ADOS-2. As no significant differences were found between participants with childhood autism and other ASDs, only one cut-off was established for Modules 1–4. The sensitivity, specificity and positive predictive value of ADOS-2-PL are high: sensitivity was over 90% (only for the “Older with some words” algorithm in the Toddler Module the sensitivity was 71% and “Aged 5 years or older” algorithm in Module 2 sensitivity was 84%), specificity was above 80% (with the exception of the Module 4 and Module 2 “Aged 5 years or older” algorithm where it was above 70%). The results support the use of ADOS-2-PL in clinical practice and scientific research.
2. Research questions has been added to the introduction. Result section has been modified.
3. The results of this study give important input to the existing literature. There is lack of studies regarding children at risk of developmental disorders, especially regarding play activity. Moreover the literature shows, that in children with ASD object play is more stereotypical and less extensive than in typically developing children. It could be expected that in this study higher severity of ASD symptoms would correlate with greater frequency of manipulating single objects. This study shows that the relationship between object play and ASD symptoms severity in children at risk of ASD is more complex and needs further studies.
Reviewer 2 Report
The article is about correlation between ASD and objects play in toddlers age 12-37 months.
We are grateful to review your article, the theme is very interesting since the observation of play section is one typical feature in our profession.
Major revision problem:
Missing the mean of ados 2 total score and subtest scores in both samples (13-18 m and 24-37 m). Lack of control neurotypical group undergone the same play section.
Minor:
Missing table with clear description of study population (paragraph 2.1). Spelling check.
We don't found any unnecessary text.
Grammar and fluidity are ok.
The tables are clear, but in paragraph 2.1 Participant...adding another table with population description and ADOS 2 total score , S.A., L.R. in both groups (13m-18-m) and (24m-37-m) could be useful.
The only limitation we highlight is the small study sample.
We hope in future to read the progression of this prospective study.
Best regards
Author Response
- The mean scores of ADOS total and subtest scores are on page 6.
- Should I replace the information in paragraph 2.1. with a table?
Reviewer 3 Report
Dear Rafał,
Thank you for allowing me to read your article!
Weak correlations may be the result of a participant group that is too homogeneous.
If the correlations are your outcome measures, I would suggest to include both neurotypical participants and children diagnosed with ASD.
Please also consider combining all children together. Maybe that will improve correlations.
Finally, I would suggest adding parent-report ASD evaluations such as ATEC and MSEC in addition to ADOS. Often, parents-reported severity may often be a better measure than ADOS since parent know their children. And these parent checklist do not add additional cost to a study.
Author Response
Unfortunately results form ADOS-2 are not available in the neurotypical group in this study. Combining all children together didn't improve the correlations. Parent-reports were not used in this study.